# Machine Learning on Ultrasound Texture Analysis Data for Characterizing of Salivary Glandular Tumors: A Feasibility Study

**DOI:** 10.3390/diagnostics14161761

**Published:** 2024-08-13

**Authors:** Li-Jen Liao, Ping-Chia Cheng, Feng-Tsan Chan

**Affiliations:** 1Department of Otolaryngology Head and Neck Surgery, Far Eastern Memorial Hospital, New Taipei 220053, Taiwan; dtent87@gmail.com; 2Biomedical Engineering Office, Far Eastern Memorial Hospital, New Taipei 220053, Taiwan; 3Department of Electrical Engineering, Yuan Ze University, Taoyuan 32000, Taiwan; 4Department of Pediatrics, Ten-Chen Hospital, Taoyuan 320680, Taiwan; ftchan@ntu.edu.tw

**Keywords:** ultrasound, texture analysis, machine learning

## Abstract

Background: Objective quantitative texture characteristics may be helpful in salivary glandular tumor differential diagnosis. This study uses machine learning (ML) to explore and validate the performance of ultrasound (US) texture features in diagnosing salivary glandular tumors. Material and methods: 122 patients with salivary glandular tumors, including 71 benign and 51 malignant tumors, are enrolled. Representative brightness mode US pictures are selected for further Gray Level Co-occurrence Matrix (GLCM) texture analysis. We use a *t*-test to test the significance and use the receiver operating characteristic curve method to find the optimal cut-point for these significant features. After splitting 80% of the data into a training set and 20% data into a testing set, we use five machine learning models, k-nearest Neighbors (kNN), Naïve Bayes, Logistic regression, Artificial Neural Networks (ANNs) and supportive vector machine (SVM), to explore and validate the performance of US GLCM texture features in diagnosing salivary glandular tumors. Results: This study includes 49 female and 73 male patients, with a mean age of 53 years old, ranging from 21 to 93. We find that six GLCM texture features (contrast, inverse difference movement, entropy, dissimilarity, inverse difference and difference entropy) are significantly different between benign and malignant tumors (*p* < 0.05). In ML, the overall accuracy rates are 74.3% (95%CI: 59.8–88.8%), 94.3% (86.6–100%), 72% (54–89%), 84% (69.5–97.3%) and 73.5% (58.7–88.4%) for kNN, Naïve Bayes, Logistic regression, a one-node ANN and SVM, respectively. Conclusions: US texture analysis with ML has potential as an objective and valuable tool to make a differential diagnosis between benign and malignant salivary gland tumors.

## 1. Introduction

For salivary gland tumors, it is difficult to have a definite diagnosis before surgical intervention [1]. Computer tomography, Magnetic Resonance Imaging and ultrasound (US) are commonly used to detect salivary gland tumors. Due to its lack of radiation exposure, point-of-care use and real-time feature, US could be used as the first-line tool to check salivary gland tumors [2,3,4]. In addition, US can be used for real-time guiding fine needle biopsy, although the diagnostic rate is reported at only around sixty percent [5]. Even with US guiding, core needle biopsy still has false negative and false positive diagnosis [6]. Therefore, the definite diagnosis is usually still dependent on surgical pathology.

High-resolution US is widely used in the preoperative evaluation for salivary tumors [7]. Previous studies have reported several subjective US features that are related to malignancy, such as calcification, loss of posterior enhancement, poor defined margin and accompanied cervical lymphadenopathy [8]. However, evaluation with US is still limited as a subjective and operator-dependent diagnostic technique.

Quantitative texture analysis of US pictures provides a more subjective assessment and is hopeful in reducing operator variations [9]. US texture analysis has been extensively applied for differentiating preterm from term fetal lungs [10], thyroid nodules [11] and chronic radiation-induced sialoadenitis [12].

Machine learning (ML) is an application of artificial intelligence, which can learn from data and may improve predictive outcomes by using data [13]. Image classification is an important application for ML, including for US pictures.

Objective quantitative texture characteristics may be helpful in salivary glandular tumor differential diagnosis. A previous study reported that texture features, including entropy and contrast, were able to differentiate benign from malignant salivary tumors [14]. However, no previous study has used ML to assess the diagnostic performance of US texture features in diagnosing salivary glandular tumors. Thus, this study aims to use ML to explore and validate the feasibility of US texture features in diagnosing salivary glandular tumors.

## 2. Materials and Methods

The inclusion criteria included patients more than 20 years old who had head and neck ultrasound examination due to salivary glandular tumors in Far Eastern Memorial Hospital. The sonograms were performed with one high-resolution 7 to 18 MHz real-time linear array transducer (Aplio MX, Toshiba, Tokyo, Japan). We recruited patients who had undergone surgery and had clear pathology. Patients without sonograms or clear pathological diagnosis were excluded. A general overview for this study is illustrated in Figure 1.

Representative brightness mode US pictures are selected for each patient (Figure 2). A total of 122 pictures are obtained for further analysis. Maximal rectangle areas within the salivary glandular tumor are delineated for Gray Level Co-occurrence Matrix (GLCM) texture analysis. We calculated eighteen texture features including angular second moment (asm), contrast, correlation, inverse difference moment (IDM), entropy, dissimilarity, inverse difference (INV), variance, cluster shade (CS), cluster prominence (CP), maximal prominence (maxpro), sum average (sumavg), sum entropy (sumenth), sum variance (sumvar), difference variance (diffvar) and difference entropy (Diffenth) and sum the average for 0, 45, 90 and 135 degrees for further comparisons [9,11,12]. We use a *t*-test to check the significance among different texture features, and select the significant predictors for further diagnostic performance assessment with ML models. After splitting 80% of the data into a training set and 20% into a testing set with the R sample function [15], we use five machine learning models, k-nearest Neighbors (kNN), Naïve Bayes, Logistic regression, Artificial Neural Networks (ANNs) and supportive vector machine (SVM), to explore and validate the performance of US GLCM texture features in diagnosing salivary glandular tumors [16]. In kNN and Naïve Bayes models, the values are normalized by subtracting the minimum value and dividing by the range. In logistic regression, we further use the ROC method to find the optimal cut-point for these significant features and split the data into categories with the cut-points. In ANN and SVM, the raw data are used in the modeling.

The GLCM texture analysis is performed using Image J (https://imagej.net/ij/index.html) [17]. All statistical analyses and ML modeling are performed by using STATA 12.0 (Stata Corp, College Station, TX 77845, USA) and R version 4.1.0 [15]. *p* values less than 0.05 were regarded as significantly different.

## 3. Results

A total of 122 patients with salivary glandular tumors, including 71 benign and 51 malignant tumors, are enrolled. There are 49 female and 73 male patients, with a mean age of 53 years old, ranging from 21 to 93. The general characteristics of the recruited patients are summarized in Table 1. There are six features that differentiate benignity from malignancy including contrast (90.2 ± 58.0 versus 129.2 ± 115.4, *p*-value = 0.03), IDM (0.28 ± 0.10 versus 0.23 ± 0.09, *p*-value = 0.02), entropy (7.01 ± 0.87 versus 7.39 ± 0.86, *p*-value = 0.04), dissimilarity (4.70 ± 1.53 versus 6.08 ± 2.72, *p*-value = 0.002), INV (0.36 ± 0.09 versus 0.32 ± 0.09, *p*-value = 0.01), and Diffenth (2.47 ± 0.31 versus 2.7 ± 0.41, *p*-value = 0.0006).

For machine learning, the overall accuracy rates are 74.3% (95%CI: 59.8–88.8%), 94.3% (86.6–100%), 72% (54–89%), 84% (69.5–97.3%) and 73.5% (58.7–88.4%) for kNN, Naïve Bayes, Logistic regression, a one-node ANN (Figure 3) and SVM, respectively. Details of diagnostic performances including sensitivity and specificity are summarized in Table 2.

## 4. Discussion

This is the first study to use ML to model the texture features for salivary glandular tumors. Our result reveals US texture analysis with ML has potential as an objective and valuable tool for the assessment of salivary gland tumors.

Dissimilarity, entropy and contrast are related to the heterogeneous content of the tumors. A previous study reported that texture features, including entropy and contrast, were able to differentiate benign from malignant salivary tumors [14]. In our study, we also find that entropy and contrast are able to differentiate benign from malignant tumors. Entropy is a quantitative measure of signal uncertainty and has been widely applied to ultrasound tissue characterization. These results mean that the malignant tumors are more heterogeneous and diverse than benign tumors.

We also include four other texture features, including the inverse difference moment, dissimilarity, inverse difference and difference entropy for ML. The inverse difference moment (IDM) is usually called homogeneity, and measures the local homogeneity of an image. The IDM feature obtains the measures of the closeness of the distribution of the GLCM elements to the GLCM diagonal. In our study, the IDM is higher for benign than malignancy (0.28 ± 0.10 versus 0.23 ± 0.09, *p*-value = 0.02).

After combining these six texture features, the diagnostic performance is 74.3% (95%CI: 59.8–88.8%), 94.3% (86.6–100%), 72% (54–89%), 84% (69.5–97.3%) and 73.5% (58.7–88.4%) for kNN, Naïve Bayes, Logistic regression, a one-node ANN (Figure 3) and SVM, respectively (Table 2). Although the performance is not perfect. Our results still support that the use of texture analysis may provide objective and quantitative information about the image pattern. Adaptation of more objective features may further increase the diagnostic performance. A computer-aided diagnostic (CAD) system for thyroid nodule sonographic evaluation has been successfully developed to assess the thyroid nodules [18,19]; in our opinion, objective US CAD for salivary gland tumors is very promising to established with ML in the future.

Artificial intelligence involves attempting to get a computer system to imitate human behavior. ML is a subset of AI techniques that attempts to apply statistical models and learning from data. ML is a field within computer science, and it differs from traditional computational approaches. In traditional computing, algorithms are sets of explicitly programmed instructions used by computers to calculate or problem solve. ML algorithms instead allow for computers to train on data inputs and use statistical analysis in order to output values that fall within a specific range. Because of this, ML facilitates computers in building models from sample data in order to automate decision-making processes based on data inputs [20].

Two of the most widely adopted ML methods are supervised learning, which trains algorithms based on data that is labeled by humans, and unsupervised learning, which provides the algorithm with no labeled data in order to allow it to find structure within its input data. This study adapts supervised learning including kNN, Naïve Bayes, Logistic regression, ANNs and SVM.

An ANN is used to develop ML systems that are based on a biological model of the brain, specifically the bioelectrical activity of the neurons in the brain. Neural networks are also called as deep learning. An ANN architecture for supervised learning could include a layer of multiple input elements, one or more hidden processing layers, and weighted connections between nodes in adjacent layers [20]. The evaluation of an ANN model (Figure 3) in our study shows that the one-node ANN model is able to correctly classify the tumor with an 84.0% accuracy rate. Due to limited data, we adapted a simple one-node ANN model; more data with multiple nodes and layers may further improve the accuracy rate.

There are some limitations in this study: First, the case number is still limited. Because this is the preliminary feasibility study, more data are needed to consolidate our findings. Second, other texture analysis methods, such as local binary pattern and multiscale features [21,22], could be used to increase the amount of data. Third, there are other forms of US pictures, such as Doppler and elastography models, that could also be adapted in future studies. Fourth, there are more ML algorithms that could be used [23].

## 5. Conclusions

US texture analysis with machine learning has potential as an objective and valuable tool to make a differential diagnosis between benign and malignant salivary gland tumors. Among the five machine learning models in our study, Naïve Bayes achieved the highest diagnostic accuracy (94.3%) using six GLCM texture features (contrast, inverse difference movement, entropy, dissimilarity, inverse difference and difference entropy).

## Figures and Tables

**Figure 1 diagnostics-14-01761-f001:**
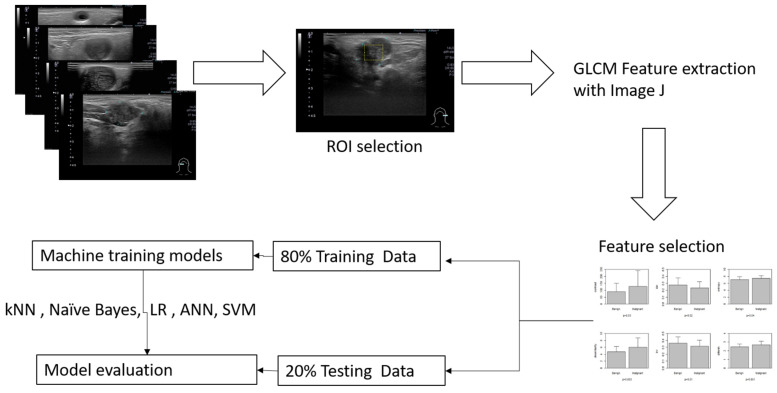
Overview of the workflow for this study, a maximal rectangle area within the salivary glandular tumor is delineated for the ROI (region of interest).

**Figure 2 diagnostics-14-01761-f002:**
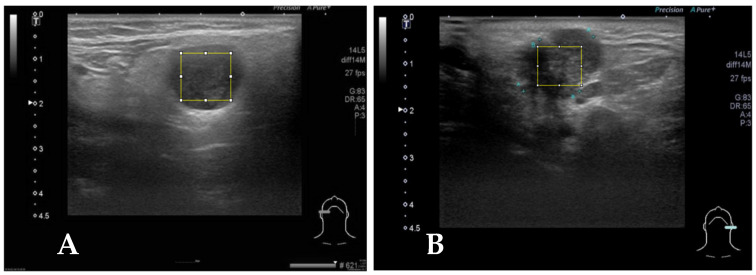
(**A**) Case 2, the square block is sampled from a right parotid tumor for GLCM texture analysis by Image J. The pathologic report reveals pleomorphic adenoma. (**B**) The square block is sampled for GLCM texture analysis from another left parotid tumor, and the pathologic report reveals mucoepidermoid carcinoma.

**Figure 3 diagnostics-14-01761-f003:**
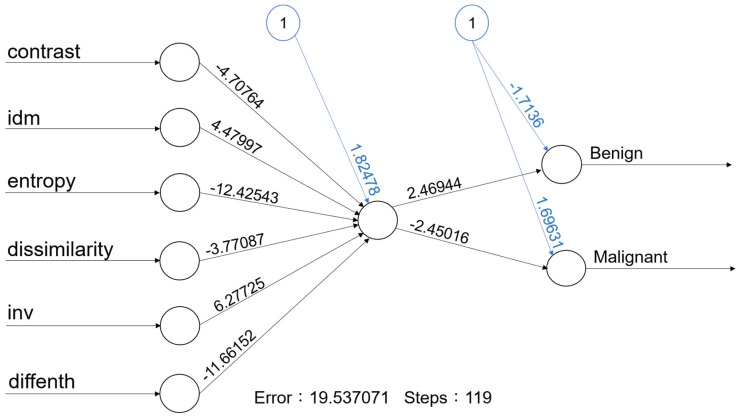
With one hidden node with 6 predictors, the accuracy rate of this ANN model is 84.0% (95% CI: 69.5–97.3%).

**Table 1 diagnostics-14-01761-t001:** Demographic and texture analysis results of recruited patients.

Characteristics	Benign	Malignant	*p*-Value
Age	50.5 ± 12.8	56.1 ± 17.8	0.06
Gender (F/M)	30/41	19/32	0.71
Size-short axis	1.58 ± 0.59	1.79 ± 0.60	0.06
Size-long axis	2.35 ± 0.95	2.51 ± 0.91	0.35
Contrast	90.2 ± 58.0	129.2 ± 115.4	0.03
IDM	0.28 ± 0.10	0.23 ± 0.09	0.02
Entropy	7.01 ± 0.87	7.39 ± 0.86	0.04
Dissimilarity	4.70 ± 1.53	6.08 ± 2.72	0.002
INV	0.36 ± 0.09	0.32 ± 0.09	0.01
Diffenth	2.47 ± 0.31	2.7 ± 0.41	0.0006
Final diagnosis	Pleomorphic adenoma (29)	Metastatic carcinoma (26)	
	Warthin’s tumor (24)	Invasive carcinoma (6)	
	Chronic sialadenitis (5)	Mucoepidermoid carcinoma (3)	
	Basal cell adenoma (4)	Acinic cell carcinoma (3)	
	Lymphoepithelial cyst (2)	Lymphoepithelial carcinoma (2)	
	Nodular fasciitis (2)	Adenoid cystic carcinoma (2)	
	Benign cyst (2)	Carcinoma ex-pleomorphic adenoma (2)	
	Epidermal cyst (1)	Adenocarcinoma (1)	
	Lipoma (1)	Diffuse large B cell lymphoma (1)	
	Reactive hyperplasia LN (1)	High-grade B cell lymphoma (1)	
		Blue round cell tumor (1)	
		Lymphoblastic lymphoma (1)	
		Squamous cell carcinoma (1)	
		Salivary ductal carcinoma (1)	

Abbreviations: IDM, inverse difference moment; INV, inverse difference; Diffenth, difference entropy; LN, lymph node.

**Table 2 diagnostics-14-01761-t002:** Summary of performance of five machine learning models with six selected texture features.

	Sensitivity	Specificity	Overall Accuracy
kNN (k = 5)	62.5 (38.8–86.2)%	84.2 (67.8–100)%	74.3 (59.8–88.8)%
Naïve Bay	88.2 (72.9–100)%	100%	94.3 (86.6–100)%
Logistic regression	75.0 (32.6–100)%	71.4 (52.1–90.8)%	72.0 (54.4–89.6)%
ANN	60.0 (29.6–90.4)%	100%	84.0 (69.5–97.3)%
SVM	87.5 (64.6–100)%	69.2 (51.5–87.0)%	73.5 (58.7–88.4)%

Abbreviations: kNN, k-nearest Neighbors; ANNs, Artificial Neural Networks; SVM, supportive vector machine.

## Data Availability

Not available due to privacy and ethical reasons.

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
