# Peer review of "Machine Learning on Ultrasound Texture Analysis Data for Characterizing of Salivary Glandular Tumors: A Feasibility Study"

_diagnostics, 2024, doi:10.3390/diagnostics14161761_

Round 1
Reviewer 1 Report
Comments and Suggestions for Authors
I read the article by Liao et al. with great interest. The topic of the article is very relevant. Ultrasound imaging is the most commonly performed diagnostic imaging modality in the practice of medicine. It is low-cost, safe, portable, and capable of realtime image acquisition and display. Ultrasound devices are developing rapidly, but the method remains operator-dependent. New promising modes are now being developed, including contrast and multimodal ultrasound, but there is a lot of information in the usual B-mode that can be used. Tremendous opportunities have arisen in the last decade as a result of exponential growth in available computational power. The problem of differential analysis of benign and malignant tumors is extremely pressing, and therefore the article will be of great interest to readers.
The study was adequately planned, relevant methods and approaches were selected. The experiments have been performed at a very high scientific level. However, while reading the article I had a few minor comments.
Materials and Methods
1. I recommend starting the Materials and Methods section with patients. Please outline the inclusion and exclusion criteria, provide the protocol and the date of the decision of the ethics committee. It would be good to add a phrase stating that informed consent is not necessary in this case due to the retrospective study.
2. Indicate in which hospital or clinic the patients were examined.
3. Please, give information on the US devices, probes, frequences etc.
3. Please, describe by what methods the final diagnoses were made.
4. Indicate how many ultrasound images were obtained for each patient and how many total ultrasound images were used for machine learning.
5. How the group was divided into the training and testing sets?
6. Specify the level of statistical significance (p < 0.05?).
7. In the caption under Fig. 1, decipher ROI (region of interest).
Results
1. Give the name of table 2.
2. In table 2, please correct Naïve Bay
Conclusions
1. Conclusions are presented in one general sentence. Please add specifics. You do not just evaluate salivary gland tumors, but make a differential diagnosis between benign and malignant neoplasms. It makes sense to conclude with a few details regarding the selected characteristics and algorithms, indicating overall accuracy.
Author Response
Materials and Methods
- I recommend starting the Materials and Methods section with patients. Please outline the inclusion and exclusion criteria, provide the protocol and the date of the decision of the ethics committee. It would be good to add a phrase stating that informed consent is not necessary in this case due to the retrospective study.
Response:Thank you very much for your suggestions. All of the revised parts are remarked in red color with underline. We add the inclusion and exclusion criteria at page 2, Line 61-66.” The inclusion criteria included patients more than 20-years old who had head and neck ultrasound examination due to salivary glandular tumors in Far Eastern Memo-rial hospital. The sonograms were performed with one high-resolution 7- to 18-MHz real-time linear-array transducer (Aplio MX, Toshiba, Tokyo, Japan). We recruited pa-tients who underwent surgery and had clear pathology. Patients without sonograms or clear pathological diagnosis were excluded.” And the protocol and the date of the decision of the ethics committee is described at page 6, Line198.” Institutional Review Board Statement: The study was conducted in accordance with the Decla-ration of Helsinki, and ap-proved by the Institutional Ethics review board of Far Eastern Memorial Hospital (IRB:112136-E, 2023.08.22).” The informed consent Statement we described at page 6, Line199-200.” Informed Consent Statement: Not applicable. Formal informed consent was waived due to ret-rospective study design.”
- Indicate in which hospital or clinic the patients were examined.
Response:Thank you very much for your suggestions. We added the hospital in materials and methods at page 2, Line 61-63:” The inclusion criteria included patients more than 20-years old who had head and neck ultrasound examination due to salivary glandular tumors in Far Eastern Memo-rial hospital.”
- Please, give information on the US devices, probes, frequences etc.
Response:Thank you very much for your suggestions. We describe the US devices, probes, frequencies information in page 2, Line 63-64:“The sonograms were performed with one high-resolution 7- to 18-MHz real-time line-ar-array transducer (Aplio MX, Toshiba, Tokyo, Japan).”
- Please, describe by what methods the final diagnoses were made.
Response:Thank you very much for your suggestions. We describe the final diagnoses were made according to pathology in page 2, Line 64-66:” We recruited patients who underwent surgery and had clear pathology. Patients without sonograms or clear pathological diagnosis were excluded.”
- Indicate how many ultrasound images were obtained for each patient and how many total ultrasound images were used for machine learning.
Response:Thank you very much for your suggestions. We describe in page 2, Line 72-73:” A representative brightness mode US pictures are selected for each patient (Figure 2). A total of 122 pictures were obtained for further analysis.”
- How the group was divided into the training and testing sets?
Response:Thank you very much for your suggestions. We describe in page 3, Line 82-83.
“After splitting 80% data into training set and 20% into testing set with the R sample function[15]…”
- Specify the level of statistical significance (p < 0.05?).
Response:Thank you very much for your suggestions. We describe in page 3, Line 92-93.
“The p values less than 0.05 were regarded as significantly different.”
- In the caption under Fig. 1, decipher ROI (region of interest).
Response:Thank you very much for your suggestions. We describe in page 2, Line 69-70.
“Figure 1. Overview of the workflow for this study, a maximal rectangle area within the salivary glandular tumor are delineated for ROI (region of interest).”
Results
- Give the name of table 2.
Response:Thank you very much for your suggestions. We revised in page 5, Line 121.
“Summary of performance of five machine learning models with six selected texture features.”
- In table 2, please correct Naïve Bay
Response: We revised us your suggestions in table 2.
Conclusions
- Conclusions are presented in one general sentence. Please add specifics. You do not just evaluate salivary gland tumors, but make a differential diagnosis between benign and malignant neoplasms. It makes sense to conclude with a few details regarding the selected characteristics and algorithms, indicating overall accuracy.
Response:Thank you very much for your suggestions. In the abstract we briefly revised in page 1, Line29-30:” Conclusion: US texture analysis with ML has potential as an objective and valuable tool to make a differential diagnosis between benign and malignant salivary gland tumors.” And in the manuscript we described as “US texture analysis with machine learning has potential as an objective and valuable tool to make a differential diagnosis between benign and malignant salivary gland tumors. Among the five machine learning models in our study, Naïve Bayes achieved the highest diagnostic accuracy (94.3%) using six GLCM texture features (contrast, inverse difference movement, entropy, dissimilarity, inverse difference and difference entropy).” (Page6, Line 180-184).

Reviewer 2 Report
Comments and Suggestions for Authors
The article "Machine Learning on Ultrasound Texture Analysis Data for Characterizing of Salivary Glandular Tumors: A Feasibility Study" is exceptionally good and I would like to congratulate the authors. The article does not require any improvements and can be published in its current form.
Author Response
Thank you for your review, we will revise our manuscript based on the suggestions of other reviewers.